# Seroprevalence of SARS-CoV-2 antibodies in Seattle, Washington: October 2019–April 2020

Denise J. McCulloch[1]*, Michael L. Jackson[2], James P. Hughes[3], Sandra Lester[4], Lisa Mills[5], Brandi Freeman[6], Mohammad Ata Ur Rasheed[4], Natalie J. Thornburg[6], Helen Y. Chu[1]

1 School of Medicine, University of Washington, Seattle, Washington, United States of America, 2 Kaiser Permanente Washington, Seattle, Washington, United States of America, 3 Department of Biostatistics, School of Public Health, University of Washington, Seattle, Washington, United States of America, 4 Synergy America, Inc., Duluth, Georgia, United States of America, 5 Eagle Global Scientific, LLC, Atlanta, Georgia, United States of America, 6 Centers for Disease Control and Prevention, Atlanta, Georgia, United States of America

* dmccull@uw.edu

## Abstract

**Data Availability Statement:** All relevant data are within the paper and its Supporting information files.

### Background

The first US case of SARS-CoV-2 infection was detected on January 20, 2020. However, some serology studies suggest SARS-CoV-2 may have been present in the United States prior to that, as early as December 2019. The extent of domestic COVID-19 detection prior to 2020 has not been well-characterized.

### Objectives

To estimate the prevalence of SARS-CoV-2 antibody among healthcare users in the greater Seattle, Washington area from October 2019 through early April 2020.

### Study design

We tested residual samples from 766 Seattle-area adults for SARS-CoV-2 antibodies utilizing an ELISA against prefusion-stabilized Spike (S) protein.

### Results

No antibody-positive samples were found between October 2, 2019 and March 13, 2020. Prevalence rose to 1.2% in late March and early April 2020.

### Conclusions

The absence of SARS-CoV-2 antibody-positive samples in October 2019 through mid-March, 2020, provides evidence against widespread circulation of COVID-19 among healthcare users in the Seattle area during that time. A small proportion of this metropolitan-area cohort had been infected with SARS-CoV-2 by spring of 2020.

**Funding:** This work was supported by the University of Washington Department of Medicine Scholars Award to HYC and by the National Institute of Allergy and Infectious Disease T32 Host Defense Training Grant (5T32AI007044-43) to DJM. Kaiser Permanente (MLJ), Synergy America Inc. (SL, MR), and Eagle Global Scientific (LM) provided support in the form of salaries for authors MLJ, SL, MR and LM, but did not have any additional role in the study design, data collection and analysis, decision to publish, or preparation of the manuscript. The specific roles of these authors are articulated in the 'author contributions' section.

**Competing interests:** The authors have read the journal's policy and the authors of this manuscript declare the following potential competing interests: Dr. Chu reported consulting with Ellume, Pfizer, The Bill and Melinda Gates Foundation, Glaxo Smith Kline, and Merck. She has received research funding from Sanofi Pasteur and Ellume, and support and reagents from Cepheid outside of the submitted work. Dr. Jackson is employed by Kaiser Permanente and reports receiving research funding from Sanofi Pasteur. Sandra Lester and Mohammad Ata Ut Rasheed were employed by Synergy America, Inc. at the time of this work. Lisa Mills is employed by Eagle Global Scientific. This does not alter our adherence to PLOS ONE policies on sharing data and materials. Denise J. McCulloch, James P. Hughes, Brandi Freeman, and Natalie J. Thornburg declare that they have no known competing financial interests or personal relationships that could have appeared to influence the work reported in this paper.

## Introduction

The greater Seattle area of western Washington State was the site of the first detected COVID-19 case in the United States on 20 January, 2020. However, serological surveys suggest that SARS-CoV-2 may have been circulating in the United States, including in Washington, as early as mid-December 2019, and that the seroprevalence of SARS-CoV-2 infection in the Western US in January 2020 was approximately 2% [1].

We aimed to estimate the prevalence of SARS-CoV-2 infection in healthcare users in the greater Seattle area from October 2019 through early April 2020 in order to further characterize the temporality of SARS-CoV-2 introductions early in the global pandemic.

## Materials and methods

Residual sera were obtained from the University of Washington Clinical Virology Laboratory. These sera were collected from inpatients and outpatients >18 years who underwent routine screening for hepatitis viruses. Sera were collected once per month, during the first week of the month, from October 2019 through January 2020, were not collected in February 2020 due to lockdowns imposed by the pandemic, and then were collected weekly beginning in March 2020. Samples were not collected from the lab on a set day of the week, but rather, were picked up when staffing needs allowed time for sample retrieval. Samples were aliquoted and frozen at -20°C until testing.

Serum samples were shipped to the Centers for Disease Control and Prevention (CDC) in Atlanta, Georgia. Sera were diluted at 1:100 and pan-IgG secondary antibody, which can detect IgM, IgG, and/or IgA was used. Samples were tested with a SARS-CoV-2-specific-enzyme linked immunosorbent assay (ELISA) using the prefusion-stabilized form of the spike (S) protein [2]. Samples were considered seropositive if the anti-SARS-CoV-2 optical density (OD) spike was equal to or greater than a cutoff of 0.4. This cutoff produced a sensitivity of 96% and a specificity of 99.3% [2]. After initial testing, sera were stored at four degrees for less than 2 days, and all positive samples underwent repeat testing with the same assay (to reduce the possibility of false positive results), and specimens were not considered positive unless they tested positive both times.

Data were analyzed in SAS 9.4 (Cary, NC). This study was approved by the Institutional Review Board of the University of Washington (STUDY #00006181). De-identified specimens and basic demographic information were obtained under a waiver of informed consent for a minimal-risk, retrospective study.

## Results

Samples from 770 participants were sent for testing; 766 were of sufficient volume for testing. Demographic data were available for 572 samples (74%). The median age of participants was 45 years (interquartile range, 32.5–60), and 50.8% were female.

All 261 samples from 2019 had OD values below the cutoff for SARS-CoV-2 antibodies (Table 1). Among 87 samples from January 8, 2020, (n = 3) 3.4% had OD values just above the cutoff on initial testing but below the cutoff upon repeat testing. Similarly, of 413 samples collected after March 1st, two sera samples collected on March 13, one collected on March 25, and one collected on April 1, 2020 had OD values just above the cutoff on initial testing but below the cutoff on repeat testing. These 4 samples from 2020 were also considered to be negative.

The first confirmed positive serum was identified from one of the 107 samples collected on March 25, 2020. Four additional positive samples were identified in March and April (Table 1). The mean OD for the 5 confirmed positive samples was 1.41 (standard deviation, 0.58). The estimated prevalence of SARS-CoV-2 antibodies in our study population during

**Table 1. Proportion of specimens testing positive for SARS-CoV-2 antibodies via ELISA assay by week of specimen collection.**

| Week of sample collection | Positive | Negative | Total | Estimated point prevalence (95% CI) |
|---|---|---|---|---|
| | n = 5 | n = 761 | n = 766 | |
| October 2, 2019 | 0 (0%) | 87 (100%) | 87 | |
| November 5, 2019 | 0 (0%) | 89 (100%) | 89 | |
| December 6, 2019 | 0 (0%) | 87 (100%) | 87 | |
| January 8, 2020 | 0 (0%) | 90 (100%) | 90 | |
| March 5, 2020 | 0 (0%) | 5 (100%) | 5 | |
| March 13, 2020 | 0 (0%) | 114 (100%) | 114 | |
| March 18, 2020 | 1 (0.92%) | 108 (99.1%) | 109 | 0.92% (0.02–5.0%) |
| March 25, 2020 | 3 (3.33%) | 87 (96.7%) | 90 | 3.33% (0.7–9.4%) |
| April 1, 2020 | 1 (1.05%) | 94 (98.95%) | 95 | 1.05% (0.03–5.7%) |
| Total for March–April samples | 5 (1.21%) | 408 (98.79%) | 413 | 1.21% (0.4–2.8%) |
| Total | 5 (0.7%) | 761 (99.3%) | 766 | |

December-January was 0 (95% CI, 0–2.1%) and from March 5–April 1, 2020 was 1.2% (95% CI, 0.4–2.8%).

## Discussion

Using samples collected from adults presenting for routine laboratory testing in the early days of the pandemic, we did not find evidence of antibodies to SARS-CoV-2 in samples collected from October 2, 2019 through March 13, 2020. Subsequently, in late March and early April, the prevalence of antibodies to SARS-CoV-2 in healthcare users in the greater Seattle area rose to 1.2%, comparable to the seroprevalence reported in Seattle-area children around the same time period [3].

In contrast to our recent study examining earlier detection of SARS-CoV-2 spike-reactive antibodies in blood donors using the same assay, which found 2% of specimens tested in December 2019 in the Pacific Northwest had spike reactivity [1], our findings did not detect antibodies in individuals accessing healthcare in the Seattle area prior to the detection of the first case on January 20. If present, it likely only infected a very small proportion of the local population, below the threshold for detection in this study. This discrepancy could be due to chance, consistent with our finding of a December-January seroprevalence of 0 with a 95% confidence interval of 0–2.1%. Neither this study or the blood donor study were designed to be cross sectional and percentages are not representative of the population. While the blood donor study suggested there could have been rare, sporadic cases of COVID-19 a few weeks earlier than expected, both studies indicate that the virus was not widely circulating before spring 2020.

The absence of earlier circulation of SARS-CoV-2 in our study is consistent with findings from two studies in Canada examining residual nasopharyngeal samples from December 2019 onward, which did not find evidence of positive SARS-CoV-2 samples prior to late February 2020 [4, 5]. By contrast, however, evidence from Europe suggests that early SARS-CoV-2 may have been circulating in France and Italy since December 2019 [6, 7].

Furthermore, our finding of a SARS-CoV-2 seroprevalence around 1% in early April is consistent with larger serosurveys in the Pacific Northwest of the United States around this time [8, 9], possibly due to rapid and widespread lockdowns imposed early in the pandemic [10]. Similar seropositivity was observed at that time in other locations with early success in controlling the spread of SARS-CoV-2, including Germany (0.97%) [11], Canada (0.7%) [12], Denmark (1.4%) [13] and Korea (0.4%) [14]. This contrasts with significantly higher SARS-CoV-2

antibody positivity in parts of the world that had more cases early on in the pandemic, including other parts of the United States (3.2–6.9%) [9], Wuhan, China (4.4%) [15], Spain (4.6%) [16], Geneva (4.8%) [17], Ariano Irpino, Italy (5.6%) [18], and northern Iran (22.2%) [19].

Our approach has some advantages over serosurveys conducted in other regions. First, ELISA has been shown to have greater specificity for the detection of SARS-CoV-2 antibodies compared to lateral flow antibody assays [20], thereby reducing the likelihood of false positives in our study. Second, all borderline or positive antibody results in our study underwent repeat testing to confirm positive results, further reducing the likelihood of our obtaining a false positive result. Third, our study included samples from October–November, 2019, a time when COVID-19 was not known to be circulating. The absence of positive test results from samples during this time period reinforces that the positive results obtained from later samples in this study were unlikely to be false positives. Finally, the collection of samples over several months allowed us to assess changes in population seroprevalence over time.

This study has several limitations. First, the relatively small sample size, involving hundreds rather than thousands of samples, limits the precision of our estimates. Second, the use of residual samples from patients tested for hepatitis selected for a group of patients who had contact with the healthcare system and may not be representative of the Seattle population as a whole. The generalizability of our findings might therefore be limited. Third, due to the use of de-identified samples, we are not able to describe the study population in detail in order to understand the representativeness of the sample. Fourth, we were unable to collect specimens in February, and SARS-CoV-2 may have been circulating in that month. Finally, with stay-at-home orders in March and April, the sample population may have varied over time. Individuals who were seeking care in March and April may have had more serious conditions than those seeking care October–January.

Despite these limitations, this study contributes important data to the limited information we have thus far on the seroprevalence of antibodies to SARS-CoV-2 early in the pandemic. First, we did not detect SARS-CoV-2 spike antibodies before March 18, suggesting that infections prior to that date were not widespread. Subsequently, it increased to 1.2% in late March/early April, consistent with the time frame of increasing confirmed COVID-19 cases in the Seattle-area at that time, corroborating the known time-frame of community spread of the virus [21]. Furthermore, the low percentage of Seattle-area adults with serologic evidence of prior SARS-CoV-2 infection indicates that, as of last spring the local population may remained susceptible to COVID-19.

Large studies of population prevalence using specimens collected from a statistically representative cross-sectional cohort were initiated last spring, and will continue to examine seroprevalence over time and across diverse geographic locations. These studies will enable more accurate estimations of infections, and mortality rates and in conjunction with vaccine rollout will allow public health officials monitor potential population immunity as we approach levels to provide herd protection.

## Supporting information

**S1 Appendix. De-identified dataset.**
(XLSX)

## Acknowledgments

**Disclaimer**: The findings and conclusions in this report are those of the authors and do not necessarily represent the official position of the Centers for Disease Control and Prevention.

Names of specific vendors, manufacturers, or products are included for public health and informational purposes; inclusion does not imply endorsement of the vendors, manufacturers, or products by the Centers for Disease Control and Prevention or the US Department of Health and Human Services.

## Author Contributions

**Conceptualization:** Michael L. Jackson, Natalie J. Thornburg, Helen Y. Chu.

**Formal analysis:** Denise J. McCulloch, James P. Hughes.

**Funding acquisition:** Helen Y. Chu.

**Investigation:** Sandra Lester, Lisa Mills, Brandi Freeman, Mohammad Ata Ur Rasheed, Natalie J. Thornburg.

**Methodology:** Sandra Lester, Lisa Mills, Brandi Freeman, Mohammad Ata Ur Rasheed, Natalie J. Thornburg.

**Resources:** Natalie J. Thornburg, Helen Y. Chu.

**Supervision:** Michael L. Jackson, James P. Hughes, Natalie J. Thornburg, Helen Y. Chu.

**Validation:** Sandra Lester, Lisa Mills, Brandi Freeman, Mohammad Ata Ur Rasheed.

**Writing – original draft:** Denise J. McCulloch.

**Writing – review & editing:** Michael L. Jackson, James P. Hughes, Natalie J. Thornburg, Helen Y. Chu.

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
