## [Decision Letter · Decision Letter 0]

28 Apr 2021

PONE-D-21-07261

Seroprevalence of SARS-CoV-2 Antibodies in Seattle, Washington: October 2019–April 2020

PLOS ONE

Dear Dr. Denise J McCulloch

Thank you for submitting your manuscript to PLOS ONE. After careful consideration, we feel that it has merit but does not fully meet PLOS ONE’s publication criteria as it currently stands. Therefore, we invite you to submit a revised version of the manuscript that addresses the points raised during the review process. Both reviewers ask you to enrich your discussion and to compare your data to other previous records and publications. Please submit your revised manuscript by 7th of May, 2021. If you will need more time than this to complete your revisions, please reply to this message or contact the journal office at plosone@plos.org. Please include the following items when submitting your revised manuscript:

We look forward to receiving your revised manuscript.

Kind regards,

Gheyath K. Nasrallah, PhD

Academic Editor

PLOS ONE

Journal Requirements:

" I have read the journal's policy and the authors of this manuscript have the following competing interests:

Dr. Chu reported consulting with Ellume, Pfizer, The Bill and Melinda Gates Foundation, Glaxo Smith Kline, and Merck. She has received research funding from Sanofi Pasteur and Ellume, and support and reagents from Cepheid outside of the submitted work.

Dr. Jackson reports receiving research funding from Sanofi Pasteur.

Denise J. McCulloch, James P. Hughes, Sandra Lester, Lisa Mills, Brandi Freeman, Mohammad Ata Ut Rasheed, and Natalie J. Thornburg have declared that no competing interests exist."

We note that one or more of the authors are employed by a commercial company: Kaiser Permanente, Synergy America, Inc, Eagle Global Scientific, LLC.

2.1. Please provide an amended Funding Statement declaring this commercial affiliation, as well as a statement regarding the Role of Funders in your study. If the funding organization did not play a role in the study design, data collection and analysis, decision to publish, or preparation of the manuscript and only provided financial support in the form of authors' salaries and/or research materials, please review your statements relating to the author contributions, and ensure you have specifically and accurately indicated the role(s) that these authors had in your study. You can update author roles in the Author Contributions section of the online submission form.

2.2. Please also provide an updated Competing Interests Statement declaring this commercial affiliation along with any other relevant declarations relating to employment, consultancy, patents, products in development, or marketed products, etc.  

Reviewers' comments:

Reviewer's Responses to Questions

**Comments to the Author**

1. Is the manuscript technically sound, and do the data support the conclusions?

Reviewer #1: Yes

Reviewer #2: Yes

2. Has the statistical analysis been performed appropriately and rigorously? 

Reviewer #1: Yes

Reviewer #2: N/A

3. Have the authors made all data underlying the findings in their manuscript fully available?

Reviewer #1: Yes

Reviewer #2: Yes

4. Is the manuscript presented in an intelligible fashion and written in standard English?

Reviewer #1: Yes

Reviewer #2: Yes

5. Review Comments to the Author

Reviewer #1: This is an interesting manuscript but it needs to be improved to be of greater interest.

It presents numerous weak points, recognized at the end of the discussion by the authors. To gain consistency it is necessary to compare its results with previous records in other nations and other continents.

The strong point of the study is that it documents (with the aforementioned limitations) the null seroprevalence prior to the pandemic outbreak in the United States.

It would be interesting to make a comparison with the sequential seroprevalence in the United States with respect to other continents

Reviewer #2: This is a short report that is effective in providing useful data for the epidemiology of SARS-CoV-2.

I would encourage (not necessary, though) the authors to include in the discussion our experience in the town of Ariano Irpino - different strategy, mass screening using serology, stronger approach useful for both screening and epidemiology study.

Please refer to https://pubmed.ncbi.nlm.nih.gov/33585828/ in the Discussion.

6. PLOS authors have the option to publish the peer review history of their article (what does this mean?). If published, this will include your full peer review and any attached files.

Reviewer #1: **Yes: **Jose F Varona

Reviewer #2: **Yes: **Carlo Buonerba

---

## [Author Response · Author response to Decision Letter 0]

7 May 2021

Response to Reviewer #1: We have added two paragraphs to the discussion to compare our results with those in other nations and continents. We have included references to data from Europe, the Middle East, and Asia in order to help put our findings into a global context. The added paragraphs are as follows:

The absence of earlier circulation of SARS-CoV-2 in our study is consistent with findings from two studies in Canada examining residual nasopharyngeal samples from December 2019 onward, which did not find evidence of positive SARS-CoV-2 samples prior to late February 2020 [4, 5]. By contrast, however, evidence from Europe suggests that early SARS-CoV-2 may have been circulating in France and Italy since December 2019 [6, 7].

Furthermore, our finding of a SARS-CoV-2 seroprevalence around 1% in early April is consistent with larger serosurveys in the Pacific Northwest of the United States around this time [8, 9], possibly due to rapid and widespread lockdowns imposed early in the pandemic [10]. Similar seropositivity was observed at that time in other locations with early success in controlling the spread of SARS-CoV-2, including Germany (0.97%) [11], Canada (0.7%) [12], Denmark (1.4%) [13] and Korea (0.4%) [14]. This contrasts with significantly higher SARS-CoV-2 antibody positivity in parts of the world that had more cases early on in the pandemic, including other parts of the United States (3.2 – 6.9%) [9], Wuhan, China (4.4%) [15], Spain (4.6%) [16], Geneva (4.8%) [17], Ariano Irpino, Italy (5.6%) [18], and northern Iran (22.2%) [19].

 Response to Reviewer #2: As noted above, we have added additional references to the discussion in order to better compare our findings to the existing literature. We have included both references suggested by Reviewer #2.

---

## [Editor Report · Decision Letter 1]

12 May 2021

Seroprevalence of SARS-CoV-2 Antibodies in Seattle, Washington: October 2019–April 2020

PONE-D-21-07261R1

Dear Dr. McCulloch,

We’re pleased to inform you that your manuscript has been judged scientifically suitable for publication and will be formally accepted for publication once it meets all outstanding technical requirements.

Kind regards,

Gheyath K. Nasrallah, PhD

Academic Editor

PLOS ONE

---

## [Editor Report · Acceptance letter]

18 May 2021

PONE-D-21-07261R1 

Seroprevalence of SARS-CoV-2 Antibodies in Seattle, Washington: October 2019–April 2020 

Dear Dr. McCulloch:

I'm pleased to inform you that your manuscript has been deemed suitable for publication in PLOS ONE. Congratulations! Your manuscript is now with our production department. 

Kind regards, 

on behalf of

Dr. Gheyath K. Nasrallah 

Academic Editor

PLOS ONE